# Living Tree Moisture Content Detection Method Based on Intelligent UHF RFID Sensors and OS-PELM

**DOI:** 10.3390/s22166287

**Published:** 2022-08-21

**Authors:** Yin Wu, Chengwu Zhang, Wenbo Liu

**Affiliations:** 1College of Information Science and Technology, Nanjing Forestry University, Nanjing 210037, China; 2College of Automation, Nanjing University of Aeronautics and Astronautics, Nanjing 211106, China

**Keywords:** moisture content (MC), living tree, radio frequency identification (RFID), machine learning, online sequential parallel extreme learning machine (OS-PELM), non-destructive sensor

## Abstract

Moisture content (MC) detection plays a vital role in the monitoring and management of living trees. Its measurement accuracy is of great significance to the progress of the forestry informatization industry. Targeting the drawbacks of high energy consumption, low practicability, and poor sustainability in the current field of living tree MC detection, this work designs and implements an ultra-high-frequency radio frequency identification (UHF RFID) sensor system based on a deep learning model, with the main goals of non-destructive testing and high-efficiency recognition. The proposed MC diagnostic system includes two passive tags which should be mounted on the trunk and one remote data processing terminal. First, the UHF reader collects information from the living trees in the forest; then, an improved online sequential parallel extreme learning machine algorithm (OS-PELM) is proposed and trained to establish a specific MC prediction model. This mechanism could self-adjust its neuron network structure according to the features of the data input. The experimental results show that, for the entire living tree dataset, the MC prediction model based on the OS-PELM algorithm can identify the MC level with a root-mean-square error (RMSE) of no more than 0.055 within a measurement range of 1.2 m. Compared with the results predicted by other algorithms, the mean absolute error (MAE) and RMSE are 0.0225 and 0.0254, respectively, which are better than the ELM and OS-ELM algorithms. Comparisons also prove that the prediction model has the advantages of high precision, strong robustness, and broad applicability. Therefore, the designed MC detection system fully meets the demand of forestry Artificial Intelligence of Things.

## 1. Introduction

In recent years, the smart utilization and protection of forest resources have become increasingly important. How to efficiently manage forest wealth is a common topic of study all over the world. Currently, making full use of living trees for the terrestrial ecosystem carbon cycle, industrial material production, wildlife habitats, and so on, are the main directions of forestry research [1]. From the microscopic point of view, the standing tree is a complex porous medium composed of wood, gas, and water within a certain structure. In particular, water is inseparable from the various physiological activities of living trees [2]. Therefore, it is necessary to regularly monitor the moisture content (MC) in trees at different growth stages to study and estimate the health status of each tree, providing a practical basis for the diagnosis of wood decay and tree irrigation. As a result, the real-time and accurate measurement of living tree MC is of great significance to the effective development of forestry informatization engineering.

There are several existing techniques for detecting the MC level of woods. The oven-drying method requires the sample to be taken back to the laboratory, and then the MC can be obtained only after complicated calculation processing. This method is quite time-consuming and laborious, and it has generally been used in calibrations [3]. The traditional resistive moisture detector is unable to measure the high MC of living trees due to the inherent defects of its working principle (i.e., only working when the MC is below the fiber saturation point) [4]. In addition, given the complex composition and regulation formula shortage, the dielectric constant measuring method is difficult to carry out in a straightforward manner in the field [5]. A novel, non-destructive MC inspection system for living trees, with low energy consumption, strong applicability, and high prediction accuracy, should be extensively explored and researched in depth.

Following the rapid development of Artificial Intelligence of Things (AIoT) technology [6,7,8], passive ultra-high-frequency radio frequency identification (UHF RFID) has attracted considerable attention for its convenience and reliability [9]. It has also been extended to the field of sensor applications, through replacing the expensive and complex sensors for monitoring environmental parameters, such as temperature, humidity, and gases [10,11]. The UHF RFID sensor has been used to obtain stable results and has shown great potential, although there is still room for further improvement in terms of the sensor accuracy, anti-interference ability, data transmission range, energy efficiency, etc. Therefore, the characteristics of the UHF RFID sensor will be significantly enhanced if advanced feature recognition and retrieval algorithms are exploited and integrated.

This article proposes a smart living tree MC detection system based on the UHF RFID technique. First, the UHF RFID sensor is installed on the bark of sample woods which have a different diameter of breast (DBH) and MC levels. Next, the received signal strength indication (RSSI) and other types of information under various environmental factors are acquired and recorded. In addition, to upgrade the detection accuracy, an innovative online sequential parallel extreme learning machine (OS-PELM) model is proposed. After that, we take the RSSI of the backscattered signal, the distance between the RFID tag and reader, the air humidity, and the DBH as joint input features; thus, a novel MC prediction mechanism based on the OS-PELM algorithm can be precisely trained and established. This online diagnostic system could efficiently alleviate the radio multipath effect, and meets the requirements for the MC monitoring of ancient and precious trees. The main contributions of this article are as follows:System architecture: A non-destructive, sustainable, and passive UHF RFID sensor for living tree MC detection has been designed, tested, and evaluated.Machine learning model: The neuron network structure of the existing sequential extreme learning machine is optimized, the idea of multiple hidden layers is introduced, and an advanced OS-PELM is proposed.MC prediction: Based on the OS-PELM algorithm, an MC prediction model for living trees is established, and its performance is proved by multi-dimensional experiments to be fully in line with the modern forestry informatization industry.Application feasibility: The design constraints of the proposed MC detection system are discussed, and its generalization ability is analyzed from the perspective of different measuring distances and MC levels.

The rest of this article is organized as follows. Section 2 introduces the related work in the field of UHF RFID wood MC detection. Section 3 presents the sensor system design and predictive model construction, including the selection of sensor tags and the operational principle of proposed algorithm. Section 4 describes the collected dataset and compares the inference performance of the OS-PELM model to demonstrate its superiority and stability. Finally, Section 5 summarizes the whole work.

## 2. Related Work

Various technical means for real-time detection of wood MC, and using UHF RFID technology for non-destructive testing in the IoT industry, have been proposed by scholars, which have a certain reference value for this article.

### 2.1. Related Literature on Wood MC Detection

In [12], Anton Fuchs et al. introduced a measurement principle for online wood moisture determination based on capacitive sensing and tested it using a laboratory prototype in a drying chamber. The experimental results show that the proposed scheme can be used to perform the preliminary detection of moisture, but the industrialization of the proposed method may still need verification. In [13], the authors proposed a capacitive multi-wavelength sensor for the gradient sensing of wood MC. By extending and modifying the previous research results, the change in positive MC gradient could be determined, which improved the accuracy of the traditional capacitive sensor. Tamme et al. proposed a new polarization-type moisture meter in [14], which improved the calibration method of the traditional resistance-type wood moisture meter. The essence of the resistance method is to use the relationship between water content and direct current conductance to measure the MC. It can also be used to detect the wood drying stage and determine the season of growing plants. The resistance method and the dielectric constant method are more flexible and are commonly used in practical measurement; however, they all need to be inserted into the trunk to a certain depth, which must cause considerable damage to the living tree. Moreover, the resistance method is only suitable for measuring target wood with an MC between 5% and 30%, as the accuracy of this detection method would decrease sharply beyond this range. Although many technical proposals are currently adopted for the MC detection of living wood, the problems of high cost, low applicability, and high energy consumption have not been reasonably resolved.

### 2.2. Related Literature on UHF RFID

In recent years, the development of passive UHF RFID has resulted in a new method for cost-efficient wireless communication. This technique does not require bulky silicon chips, instead using electromagnetic signatures to encode data, reducing the cost of RFID tags [15]. In addition, it has the characteristics of backscatter transmission and non-insertion detection, which provides an innovative solution for humidity-sensing research in various areas. In [16], Joan Melià-Seguí et al. deployed multiple RFID tags inside a car to measure the RSSI and RF phases. They found the sensitivity of the RSSI and RF phases to the water, and creatively applied RFID technology to humidity detection in the interior space of cars. Pichorim et al. demonstrated that RFID chips have the ability to detect soil wetness, and they carried out a preliminary experiment to certify this [17], but they did not build a complete MC prediction model. Rafael V. et al. proposed a sensing system for monitoring soil moisture by using UHF RFID tags buried in the soil, and they established an MC prediction model based on neural networks. The experimental results proved the feasibility of their idea [18]. However, due to the insufficient generality and robustness of the neural network model, their research results still have room for improvement. At present, there are few traditional algorithms proposed to solve the problem of MC retrieval using the information obtained from RFID sensors [19,20].

Thus far, existing studies have demonstrated the application prospects of UHF RFID technology in the field of MC detection. However, to the best of our knowledge, there is no relevant research report using this method to detect the MC of living wood. This article focuses on moisture monitoring sensors to provide a state-of-the-art solution for the development of Internet of Things in forestry.

## 3. Materials and Methods

This section introduces the MC prediction system, including an efficient machine learning model. It is divided into several parts: sensing principle, system structure, OS-PELM algorithm design, and implementation scheme.

### 3.1. Sensor Description

The operational object of this MC detection mechanism is an UHF RFID system capable of receiving tag signals, including devices such as RFID tags, an antenna, and a reader, as shown in Figure 1. In addition to reliably verifying the ID number of each tag, the reader accurately returns an RSSI value of the backscatter signal. The system also includes a computing terminal responsible for processing the signal information and analyzing the MC condition. Note that the practical MC level of living wood is calibrated by a MD914 high-precision wood moisture meter in the testing stage [21]. At the same time, the nearby air humidity, the DBH of the wood, the distance between tags, and the reader’s antenna are collected and recorded as supplementary parameters to construct a deep learning prediction dataset as accurately as possible.

Here, the reader adopted was the off-the-shelf UHF RKF918, a full-featured micro reader which can work in the frequency band of 840–960 MHz. The UHF-R200 module equips a multi-tag anti-collision architecture and supports the EPC Gen2 protocol [22]. It can read and write RFID tags up to 20 m. The antenna used in the experiment was a RYT-280 8.5 dBi circularly polarized antenna, which adopts a dual-feed circularly polarized design to meet the requirements of small size, high gain, and low voltage standing wave ratio.

In order to improve the testing efficiency of this measuring scheme in practical application scenarios, we selected six types of the most common UHF RFID tags in the academic field [23], as shown in Figure 2. The sensitivity of these tags to the reading distance and wood MC was examined and compared through the following procedures in advance. The groups of six different tags are numbered as label 1 to label 6, in sequence. Each sample has two paired tags, one for reference and the other for real sensing: the sensing tag should be placed on the opposite side of the trunk, far from antenna, while the close-by control tag is used for comparison, directly facing the reader. When the radio wave travels through the wood, it declines based on some related factors (especially the water content), so the RSSI values of two tags should differ greatly, making full use of their interaction relationship.

In order to confirm the discrimination of RSSI values received by the reader, a custom antenna is temporary placed one meter away from each group. The corresponding measurement results are shown in Figure 3.

For using the RSSI value to establish an MC prediction model more accurately, it is required that the changes in numerical value should be more sensitive to the distance between the antenna and tags. That is, the measured RSSI value of the sensing tag and the control tag should have a larger disparity. It is apparent that tag 1 is the most sensitive RFID during the signal propagation process (the difference between the measured value and the average control value is 19 dBm). Therefore, the proposed MC diagnosis sensor is implemented by the square four-corner label, as shown in Figure 4. The tag uses transparent polyvinyl chloride as a waterproof coating and can be attached on the trunk safely.

### 3.2. UHF RFID Technology

The sensing process of UHF RFID technology includes two main parts: one is the signal transmission between the tag chip and the reader, and the other is the information communication between the RFID reader and the processing computer. As shown in Figure 5, the communication process between the UHF RFID tag and the reader adopts the backscatter modulation technique: it basically uses the influence of the sensing object’s variation in the backscattering signal characteristics of the RFID system to sense the change in the target state [24].

The RFID reader can provide some important information coefficients used as sensing parameters in wood MC detection scenarios, such as RSSI, Phase, and Doppler frequency shift. The RSSI from a tag is determined by a number of factors, including the tag design, its distance to the reader, and the environment (i.e., the presence of objects affecting the signal propagation). It is relatively easier to obtain the RSSI in a complex forest environment, and to reflect the characteristics of the transmission medium between the tag and reader [25].

Based on free space, polarization matching, no loss, and port matching, the signal power received by the receiving UHF RFID antenna is:(1)$$P_{r}=\frac{P_{tx}G_{tx}}{{4\pi d}^{2}}\cdot \frac{G_{tag}\lambda^{2}}{4\pi },$$

The meanings of the parameters are shown below, in Table 1.

According to Equation (1), after determining *P_r_*, *G_tag_*, *P_tx_*, and *G_tx_*, the maximum readable distance, *dm*, of the tag is:(2)$$dm=\frac{\lambda }{4\pi }\sqrt{\frac{\;P_{tx}G_{tag}G_{tx}}{P_{r}}},$$

In the actual process of applying the RFID sensing system, the backscattered power of the tag is an important parameter. Additionally, based on Equation (1), the backscattered power received by the reader can be calculated by the following equation [26]:(3)$$P_{tag-back}{=P}_{tx}{\left(\frac{G_{tag}G_{tx}\lambda^{2}}{{\left(4\pi d\right)}^{2}}\right)}^{2}\tau \left(\Delta \right).$$

The transmission power of the reference signal in the free space is used as the theoretical reference of this paper, and a prediction scheme is proposed according to Equation (3). The magnitude of the backscattered power *P_tag-back_* depends on several factors: when the tag is fixed in the practical working site, all parameters except Δ can be considered as constant. In this study, the coefficient Δ reflects the change in the dielectric properties of the transmission medium between the tag and the reader, which is mainly affected by the moisture content of the wood. Therefore, RSSI can be used to infer the MC level of living wood.

### 3.3. OS-PELM Algorithm

#### 3.3.1. Online Sequential Parallel Extreme Learning Machine

The extreme learning machine (ELM) is an important branch of the single hidden layer feedforward neural network, in which the parameters of input weights and hidden layer bias can be initialized randomly, and only the output weights need to be optimized [27,28]. The parallel extreme learning machine (PELM) is proposed on the basis of ELM, and the advantage is that the input layer is simultaneously mapped to various hidden layers with different activation functions, which can concurrently learn the heterogeneous features from the inputs.

Figure 6 shows the structure of the proposed OS-PELM model. The algorithm has the following three advantages:In the input layer, two single hidden layers and a single-layer feedforward neural network can transmit information simultaneously.The original input information in the middle can also be directly fed to the output layer, which enhances the linear data processing.Neurons with the property of online sequential learning are introduced. The current prediction is largely dependent on the features of the current training data, while the former data are discarded gradually after the last learning period.

The OS-PELM learning process is basically divided into two parts: one is the initial phase, in which initial output weights could be obtained through a small number of samples; the other is the online learning phase, where the data blocks should be imported into the network in order, and the output weights can then be updated. Here, we consider *N* arbitrary samples {(***x_t_***, ***y_t_***), *t* = 1, 2, …, *N*}, where ***x****_t_*= [***x*_1*t*_**, ***x*_2*t*_**, ***x*_3*t*_**, …, ***x_nt_***]^T^ ∈ **R**^n^ refers to the *n*-dimensional feature vector of the *t-*th sample, and ***y_t_*** is the output vector of the *t*-th sample. The weights between the input layer and two parallel hidden layers are defined as ***ρ*** and ***φ***; *b_1_* and *b_2_* refer to the biases of hidden layers; ***β*_1_** and ***β*_2_** are the output weights between the hidden layers and output layer; and ***ω*** is a matrix that has the weight values directly connecting the input layer and output layer. *h*(·), *g*(·) and *f*(·) represent the activation functions of the hidden layers and output layer, respectively. Note that the input data are divided into two groups (***x*_1_**-***x_m_***) and (***x_m+_*_1_**-***x_n_***), which correspond to the system-dependent parameters (the distance from reader to tag, RSSI value of the control tag, and RSSI value of the sensing tag) and environment-dependent parameters (DBH and the air humidity). The aim is to analyze the MC attributes of living trees more accurately based on the features of different kinds of parameters. Then, the output ***y_T_*** of OS-PELM is as follows:(4)$$y_{T}^{\prime }=\overset{z1}{\underset{k=1}{\sum }}\beta_{1k}\cdot \;h\left({\rho \;x}_{1}^{m}{+b}_{1k}\right),\;$$
(5)$$y_{T}^{\prime \prime }=\overset{z2}{\underset{k=1}{\sum }}\beta_{2k}\;\cdot \;g\left({\varphi \;x}_{m+1}^{n}{+b}_{2k}\right),$$
(6)$$y_{T}^{\prime \prime \prime }{=\omega \cdot \;x}_{1}^{n},$$
(7)$$y_{T}=f\;\left[y_{T}^{\prime }{+y}_{T}^{\prime \prime }{+y}_{T}^{\prime \prime \prime }\right]=f\;[\overset{z1}{\underset{k=1}{\sum }}\beta_{1k}\cdot \;h\left({\rho \;x}_{1}^{m}{+b}_{1k}\right)+\overset{z2}{\underset{k=1}{\sum }}\beta_{2k}\;\cdot \;g\left({\varphi \;x}_{m+1}^{n}{+b}_{2k}\right){+\omega \cdot \;x}_{1}^{n}],$$

Here, for strict calculation purposes, ${\;x}_{1}^{m}$ and${\;x}_{m+1}^{n}$ are expanded to *n* dimensions with zero filling in the blanks. Thus, Equation (7) can be rewritten as:(8)$$Y_{T}{=f\;(\beta }_{1}{H+\beta }_{2}G+\omega \;\chi )=f\left({[\beta }_{1}\beta_{2}\omega ]\left[\begin{array}{c} H\\ G\\ \chi \end{array}\right]\right).$$
where the calculation rules of ***H*** and ***G*** are as follows:(9)$$H_{i}=h\left({\rho \chi }_{i}{+b}_{1}\right),$$
(10)$$G_{i}=g\left(\varphi \chi_{i}+b_{2}\right),$$
(11)$$H={\left[H_{1},\;H_{2},\ldots ,\;H_{n}\right]}^{T},$$
(12)$$G={\left[G_{1},\;G_{2},\ldots ,\;G_{n}\right]}^{T}.$$

In addition, the objective function of PELM is equivalent to the optimization problem of ELM. The output weight ***β*** can be determined by the least-norm least-squares solution.
(13)$$\beta^{\left(0\right)}={\left[\begin{array}{c} H\\ G\\ X \end{array}\right]}_{0}^{+}{\cdot Y}_{T}^{0}={\left({\left[\begin{array}{c} H\\ G\\ X \end{array}\right]}_{0}^{T}{\left[\begin{array}{c} H\\ G\\ X \end{array}\right]}_{0}\right)}^{-1}\cdot {\left[\begin{array}{c} H\\ G\\ X \end{array}\right]}_{0}^{T}{\cdot Y}_{T}^{0},$$

After entering the online learning stage, in the *k +* 1-th (*k* = 0, 1, ‧‧) learning stage, the newly arrived *N_k+_*_1_ samples are input into the proposed model, and ***β***^(k+1)^ should be recalculated as:(14)$$\beta^{\left(k+1\right)}{=\beta }^{\left(k\right)}{+F}_{k+1}{\left[\begin{array}{c} H\\ G\\ X \end{array}\right]}^{T}\left(Y_{k+1}-\left[\begin{array}{c} H\\ G\\ X \end{array}\right]{\cdot \beta }^{\left(k\right)}\right),$$
where the determination of ***F_k+_*_1_** is as follows:(15)$$F_{k+1}{=F}_{k\;}-{\;F}_{k}{\left[\begin{array}{c} H\\ G\\ X \end{array}\right]}_{k+1}^{T}{\left(I+{\left[\begin{array}{c} H\\ G\\ X \end{array}\right]}_{k+1}{\cdot F}_{k}{\left[\begin{array}{c} H\\ G\\ X \end{array}\right]}_{k+1}^{T}\right)}^{-1}{\left[\begin{array}{c} H\\ G\\ X \end{array}\right]}_{k+1}{\cdot F}_{k},$$
(16)$$F_{0}=\;{\left({\left[\begin{array}{c} H\\ G\\ X \end{array}\right]}_{0}^{T}{\left[\begin{array}{c} H\\ G\\ X \end{array}\right]}_{0}\right)}^{-1}.$$

#### 3.3.2. Structural Optimization of OS-PELM

Nevertheless, the OS-PELM architecture still has some shortcomings: due to the online sequence module, the pre-given network structure may no longer be able to achieve the expected prediction accuracy. Therefore, an optimization skill is introduced as described below.

When OS-PELM is in the online learning stage, the hidden layer neurons are divided or merged according to their contribution to the prediction results, and the network structure is adjusted to highlight the time effectiveness of the training network and ensure its stability and accuracy. Here, the degree of contribution is defined as:(17)$$C_{i}^{\prime \;}=\left(\frac{\beta_{i}\lambda_{i}}{y_{T}^{\prime }}\right)\;\cdot \left[\frac{y_{T}^{\prime }}{y_{T}^{\prime }{+y}_{T}^{\prime \prime }{+y}_{T}^{\prime \prime \prime }}\right],$$
(18)$$C_{j}^{\prime }=\left(\frac{\beta_{j}\lambda_{j}}{y_{T}^{\prime }}\right)\;\cdot \left[\frac{y_{T}^{\prime }}{y_{T}^{\prime }{+y}_{T}^{\prime \prime }{+y}_{T}^{\prime \prime \prime }}\right],$$
where $C_{i}^{\prime }\;$and $C_{j}^{\prime }$ represent the degrees of two hidden layer neurons, respectively. ***β_i_*** and ***β_j_*** are the output weights of the *i-*th and *j-*th neurons. *λ_i_* and *λ_j_* are the output values. Whenever the degree of a neuron exceeds the highest threshold (*C_Hth_*) or jumps below the lowest threshold (*C_Lth_*), this original neuron is split into several new neurons or merged with other nearby neurons. For the weights between the input layer and the two parallel hidden layers, ***ρ*** and ***φ***, the bias of hidden layers, ***b***, and the output weights between hidden layers and the output layer, ***β***, specific adjustment rules are set, as follows:(19)$$\{\begin{array}{c} \;\rho_{new}=\rho_{original}\\ \;\varphi_{new}={\varphi_{original}}^{\prime } \end{array}$$
(20)$$\{\begin{array}{c} b_{new}^{\prime }{=\Omega \;\times \;b}_{original}\\ {\;b}_{new}^{\prime \prime }=\left(1-\;\Omega \right){\;\times \;}_{{b_{original}}^{\prime }} \end{array}$$
(21)$$\{\begin{array}{c} \beta_{new}^{\prime }{=\Omega \;\times \;\beta }_{original}\\ {\;\beta }_{new}^{\prime }=\left(1-\;\Omega \right){\;\times \;}_{{\beta_{original}}^{\prime }} \end{array}$$

Here, the input weights of divided neurons remain the same as before, and the output parameters must be multiplied by a variation factor **Ω**. Meanwhile, the merged neurons are removed and the previous neuron’s output weight ***β*** should be as shown in Equation (22):(22)$${\beta =\beta }_{original}+\frac{\beta_{delete}\;\lambda_{delete}}{\beta_{original}\;\lambda_{original}}\beta_{delete}.$$

Note that ***β_delete_***, ***β_original_***, and ***β_new_*** are the output weights of the deleted neuron, the original neighboring neuron, and the newly merged neuron, respectively. In summary, the whole working procedure of OS-PELM is given in Algorithm 1.
**Algorithm 1** Proposed OS-PELM**Input:** A training set {(xt,yt)}i=1N, activation function *h*(*x*), *g*(*x*), *f*(*x*), two hidden layer node number Z1, Z2, thresholds *C_Hth_*, *C_Lth_*.**Output:** The actual predicted value *Y_T_*.1: **Initialize**: Randomly generate weights ***ρ***, ***φ***, ***ω***, biases ***b*_1*t*_**, ***b*_2*t*_**.2: Calculate the initial output weight matrix β(0) by using Equations (9)–(13).3: Calculate the output YT0 and corresponding standard deviation with *Y_T_*.4: **Online adjust**: Newly arrived samples should be inputted into the prediction model, relevant parameters would be regulated by using Equations (14)–(16).5: Compute the contribution degree for all hidden layer nodes with Equations (17) and (18).6: **if** one neuron needs to be divided **then**7:   Calculate new neurons’ parameters by Equations (19)–(21);8: **else if** one neuron needs to be merged9:   Calculate new neuron’ output weight by Equation (22);10: **end if**11: **Predict**: Adjustment has been completed. Actual prediction should be carried on with the next data during the same month. However, when a new time point comes, the algorithm needs to be recycled from line 5.12: **END**

### 3.4. Proposed Scheme

The establishment of the entire model is shown in Figure 7. The basic principle of the MC prediction model based on deep learning is to map the input features (RSSI, measuring distance, air humidity, and DBH) to the output-specific MC value. We aim to employ an appropriate intelligent model to find the optimal computational mechanism for the MC prediction scheme. Considering the time-varying characteristics of the receiving signal parameters and environmental conditions, we develop a novel online sequence learning algorithm, which can learn and update the model from the newly received data without tedious retraining. Only then can the diagnostic system meet the needs of real-time UHF RFID sensing.

For our proposed OS-PELM algorithm, the strategy of online network structure adjustment is adopted, which greatly improves the generalization ability of the prediction model and effectively reduces the occurrence of overfitting. RSSI values of different living woods are read through the UHF RFID sensors, and the DBH, distance, air humidity, and real MC values of the trees, as measured by standard instruments, are used as input parameters. These datasets are taken as the offline training samples. After the preprocessing, these datasets are used to train the OS-PELM algorithm, in order to obtain the optimal final prediction model. Finally, the diagnostic model is applied to predict the MC level of a living tree, and the effectiveness of this proposed mechanism is verified by comparisons with diverse methods.

## 4. Results and Discussion

In this section, we introduce the experimental scenario and the data acquisition process. We also thoroughly evaluate and validate the proposed wood MC detection system.

### 4.1. Experimental Scenario and Implementation

Figure 8a shows the equipment distribution of the laboratory test scenario. Two labels are attached to the reverse side of the wood sample and placed at the same height. The two practical installed control and measuring labels are shown in Figure 8b. The aim of the control tag is to obtain a raw RSSI value free of moisture and wood medium effects, which can be used as a benchmark to facilitate the classification of the experimental samples during measurement. The computing terminal is configured with 2.66 GHz CPU, 16 GB RAM and the Windows 10 system.

As for the examination in the dynamic living tree scenario, we selected different kinds of standing trees for repeated measurements in the experimental forestry farm of Nanjing Forestry University from 20 October to 30 December 2021, as shown in Figure 9a. As above, each pair of RSSI values was recorded 100 times continuously within 100 s, and the average number was taken for the subsequent computation of numerical data. In the meantime, the straight-line distance between the reader and tags, the real MC value of the trunk measured by the MD914, as shown in Figure 9b, the DBH of the trunk, and the air humidity were all stored as the model training dataset. All parameters mentioned were collected periodically at 15 min intervals.

Figure 10 shows two RSSI numerical curves with a varying distance from tag to reader, which was tested in the lab. Red values are the results of the control label (without moisture interference), while black values are those of the measurement label (penetrating through the wood). Compared with the control value, the presence of the trunk severely affects the backscattered radio signal propagation, and this effect becomes more obvious with the increase in transmission distance. The value of RSSI fluctuates during measurement, and the figure only lists the median of several groups of RSSI measurement values. Nonetheless, the RSSI variation exhibits a complex correlation with distance, which requires in-depth analysis by an advanced methodology.

Table 2 presents the details of several parameters, in some cases as examples. Since the RSSI value is unstable during actual measurement, this instability is reflected in the fluctuation of the RSSI value upward and downward by 1 dBm. The values listed in the table are the averages of 100 consecutive measurements within 60 s. As can be seen from Table 2, for the same wood specimen, a change in the MC causes variation in the RSSI. When the MC increases from 14.8% to 25.6%, its average RSSI value changes from −10.05 dBm to −13.21 dBm. Additionally, the measuring distance also causes fluctuation in the RSSI: if the distance is improved by 0.2 m, the average RSSI changes from −18.88 dBm to −27.34 dBm. Moreover, the DBH and air humidity also have an influence on RFID signal transmission. If the MC level (around 15%) and the measurement distance (0.3 m) are basically the same, the RSSI measuring values of two samples under different DBH and air humidity are −10 dBm and −8 dBm, respectively. The change in this value is subtle, but it shows potential for building predictive models with it.

As a result of the above findings, we can infer that these data features all provide insights into the wood MC measurements. However, the correlation between the influences of the various measurements changes in different scenarios. Therefore, an intelligent prediction model based on a machine learning algorithm is desirable for the accurate measurement of wood MC.

### 4.2. Machine Learning Results

It should be noted that the RSSI values used in all experiments below are the average of 100 consecutive readings within 60 s. To verify the performance of the proposed diagnostic system, sample woods with moisture contents of 5%, 10%, 15%, 20%, 25%, 30%, 35%, 40%, and 45% (margin of error was ±1%), are obtained in the laboratory. In addition, the basis for determining the MC level in the dataset and the sample manufacturing method are processed with reference to our previous work [29], and a total of 1000 sets of sample data are obtained from the laboratory. These groups are randomly divided into three parts according to the ratio of 2:6:2, namely 200 groups of initialization sets, 600 groups of training sets, and 200 groups of testing sets. The MC identification results of the target wood by the OS-PELM algorithm, the ELM algorithm, and the OS-ELM algorithm are computed and compared. Here, the ELM algorithm and OS-ELM algorithm are condensed versions of the OS-PELM algorithm, lacking the online sequence data updating and parallel neuron adjustment, respectively.

Note that to improve the robustness of the model and achieve high prediction accuracy, all original data sets should be normalized to values within [0, 1]:(23)$$x^{*}=\frac{x_{i}-x_{min}}{x_{max}-{\;x}_{min}}\;(i=1,\;2,\;\text{…},\;N).\;$$
where *x_min_* and *x_max_* are the minimum and maximum values of the input data of a certain period, and *x** and *x_i_* are the data obtained after normalization and the original input data, respectively.

Meanwhile, the hidden layer functions *h*(‧), *g*(‧), and *f*(‧) are selected with ReLU, Leaky ReLU, and Maxout. Z1 and Z2 are first set to 10 and 18. *C_Hth_* is twice the average contribution degree, and *C_Lth_* is a minimum of {0.05, 1/(Z1 + Z2)}. In the setting of the hidden layer nodes of the extreme learning machine, in order to prevent the occurrence of overfitting and underfitting, a large number of experiments were carried out and the model topology adjusted, following which the number of hidden layer nodes of the extreme learning machine was set to 64.

The prediction results of three methods were calculated and are displayed in Figure 11. For the experiments using nine different sample boxwoods with an MC range from 5% to 45%, the prediction performance of the OS-PELM means is significantly better than the outcomes of the ELM and OS-ELM algorithms. In addition, the average error between the diagnosis output and the ground truth is lower than 4.4%. Hence, the laboratory detecting system shows a high precision and adequate data stability.

To verify the reliability of the proposed system for MC detection in living woods, the corresponding data from sample trees (metasequoia, poplar, pine, and beech) in the artificial forest station were collected for analysis, with MC levels of 45.9%, 26.1%, 34.7%, and 40.4%. The comparative experimental results are shown below in Figure 12. The system still shows good diagnostic performance for the online dynamic data, and the relative error bar indicates that the detection accuracy is higher than 97.1%. It is apparent that the OS-PELM can accurately identify the MC status for both coniferous and broadleaved woods, which proves that this diagnostic model has strong generalization ability and good robustness.

### 4.3. Performance Evaluation

Here, the mean absolute error (MAE), root-mean-square error (RMSE), and goodness of fit (R^2^ coefficient of determination) are chosen as the evaluation criteria for the experimental results. MAE and RMSE can be calculated by the following formulas:(24)$$\mathrm{MAE}=\frac{1\;}{m}\sum_{i=1}^{m}\left|h\left(x_{i}\right)-y_{i}\right|,$$
(25)$$\mathrm{RMSE}\left(X,h\right)=\sqrt{\;\;\frac{1\;}{m}\sum_{i=1}^{m}{\left(h\left(x_{i}\right)-y_{i}\right)}^{2}}.$$
where *h*(*x_i_*) represents the actual value of the *i*-th group of data, and *y_i_* represents the predicted value in the *i-*th group. Goodness of fit refers to the degree of fit of the regression line to the observed values, and the statistic to measure the goodness of fit is the coefficient of determination R^2^. The value range of R^2^ is [0, 1]. The closer the value of R^2^ is to 1, the better the fit of the regression model to the observed values; the closer the value of R^2^ is to 0, the worse the fit of the model to the observed values.

Then, we conducted evaluation dataset construction on the standing pine for 24 h. All of the relevant parameters were collected, and the three mechanisms operated 25 times each, as shown in Figure 13. For every test, the prediction accuracy of OS-PELM is higher than that of the other two algorithms. Moreover, the MAE and RMSE of the OS-PELM algorithm for MC prediction do not exceed 0.1, and the R^2^ coefficient is maintained above 0.9, which strongly proves the accuracy and reliability of the method.

OS-ELM adds an adaptive online learning stage to the ELM algorithm, so the prediction accuracy is higher, and the learning rate is faster, but the improvement is not noticeable. Due to the double hidden layer structure, the OS-PELM model can better deal with the input data features. Meanwhile, it adopts the strategy of online network adjustment, which greatly improves the efficiency and stability. For the same dataset, a specific performance comparison of the three MC prediction models is given in Table 3.

It can be seen that the MAE of our proposed wood MC prediction model is 0.0225, which is much lower than the 0.048 and 0.0488 for the other two models, respectively, and the RMSE is 0.0254, which is nearly half of the other two algorithms’ results. The R^2^ coefficient of the three prediction methods increased from 0.8086 to 0.9612, which matches the added function modules. For the analysis of training time, due to the structure of multiple hidden layers, OS-PELM runs slightly longer than the other two models, but this does not affect its detection effectiveness.

### 4.4. Robustness Analysis

In order to evaluate the robustness of the proposed wood MC diagnostic system, we carried out a comparative experiment on the variation in interval distance between the reader antenna and the testing wood. The subjects of the experiment included wood with a defined moisture content prepared in the laboratory and standing wood outdoors. The evaluation indicators still used RMSE and R^2^. We deployed the antenna and RF tags based on the above steps to ensure that the tags and antenna were placed on the same horizontal line. Then, we selected three types of sample wood with different MC levels (approximately 10%, 25%, and 40%) to test the prediction performance.

The corresponding results are shown in Figure 14a. As the distance between the antenna and trunk increases, the prediction accuracy decreases, especially when the distance exceeds 0.4 m. The main reason is that the proportion of RF signals that can be transmitted through the wood is reduced. When the interval distance is within 1 m range, the RMSE of the proposed OS-PELM prediction model is less than 0.04.

In addition, we selected one standing cypress (0.22 m DBH) and took its three MC values (31.5%, 38.0%, and 40.5%) as the objects of comparison to verify the applicability of the proposed system. The results are displayed in Figure 14b. For the cypress samples, significant detection accuracy within a 0.5 m range was obtained. For the trees with larger DBH, higher-quality RSSI signals should be obtained in a closer range. On the other hand, when the distance exceeds 0.5 m, a considerable part of the RF signal would be diluted by environmental factors, seriously affecting the prediction results. Meanwhile, the two results also indicate that the living tree MC could be diagnosed accurately under certain measurement settings.

In order to further verify the proposed algorithm, we compared the performance of the above three algorithms again based on the standing cypress data. Figure 15 reveals that the prediction accuracy of the three models decreases rapidly when the distance increases, but OS-PELM is apparently more stable and reliable. In particular, when the distance is 0.8 m, its RMSE can be controlled at 0.03, while the results of ELM and OS-ELM are as high as 0.06, far exceeding the upper error limit of MC measurement. Therefore, the proposed MC prediction model is highly efficient and effective.

### 4.5. Comparison with Similar Processing Methods

To evaluate the overall performance of the UHF RFID living tree MC diagnostic system, we used four other noted data processing algorithms to deal with the previous experiment results: (1) Differential evolution algorithm-based OSELM (DE-OSELM) [30]. Here, DE is used to solve the problem of input weight and the random selection of the hidden layer bias. Sigmoid is chosen as the activation function, the number of neurons in the hidden layers is 25, and the size of the dataset to be learned in each step is 40. (2) Information perception weight and error prediction-based OSELM (WE-OSELM) [31]. This model improves the forgetfulness factor and weighted error compensation coefficient to enhance the prediction accuracy. The number of neurons in the hidden layer is 200, the adjacent area of input data width is 6, and the activation function is also Sigmoid. (3) Gravitation search algorithm-based PELM (GSA-PELM) [32]. Here, GSA is used to optimize the hidden layer threshold and input weight of the PELM model. The number of neurons in the hidden layer is 42, and its parameter optimization range is [−1, 1]. (4) Online least square PELM (OLSPELM) [33]. The weights and thresholds of this model are determined by two least squares methods, the maximum neuron number in each hidden layer is set as 50, and the input and output attributes of data are both normalized to [0.01, 1]. Then, the detection accuracy of each method on the testing set is recorded, and the results are compared in Figure 16.

We adopted the dataset of standing cypress with four MC levels, with the reader antenna placed 0.4 m away from the trunk. As the results show, the *RMSE* of our proposed model is consistently lower than 0.0275, being at least one order of magnitude more accurate than other models. Additionally, the histogram value is smooth and steady, and does not change observably with the MC status. Therefore, the generalization ability and prediction accuracy of the OS-PELM model are much better than those of the other four models.

## 5. Conclusions

In this article, we propose an ultra-low-cost MC sensor prototype based on standard UHF RFID tags. First, in the hardware structure design, we select the most suitable measurement shape by comparing the RSSI performances of various tag shapes. Then, a data acquisition system is built efficiently, and the RSSI values and other corresponding environmental parameters are collected by fixed-cycle operations. After that, for the improvement of the existing recognition mechanism, an OS-PELM architecture is proposed to optimize the network structure of the hidden layers, adopting the strategy of online neuron adjustment to regulate the model parameters. The experimental results on the standing tree show that the MAE between the predicted value and the ground truth is 0.0224, the RMSE is 0.0254, and the R^2^ coefficient is 0.9612. Comparative experiments indicate that the performance of our model is better than that of the DE-OSELM, WE-OSELM, GSA-PELM, and OLSPELM models. Hence, it is proved that the proposed MC diagnostic system is accurate and robust in terms of monitoring performance and is suitable for use in the practical Internet of Things industry.

The UHF RFID technology can provide innovative solutions to some of the problems encountered in forestry measurements. Our further research aim is to detect the distribution of moisture content inside the trunk with high accuracy by arranging multiple RFID tags.

## Figures and Tables

**Figure 1 sensors-22-06287-f001:**
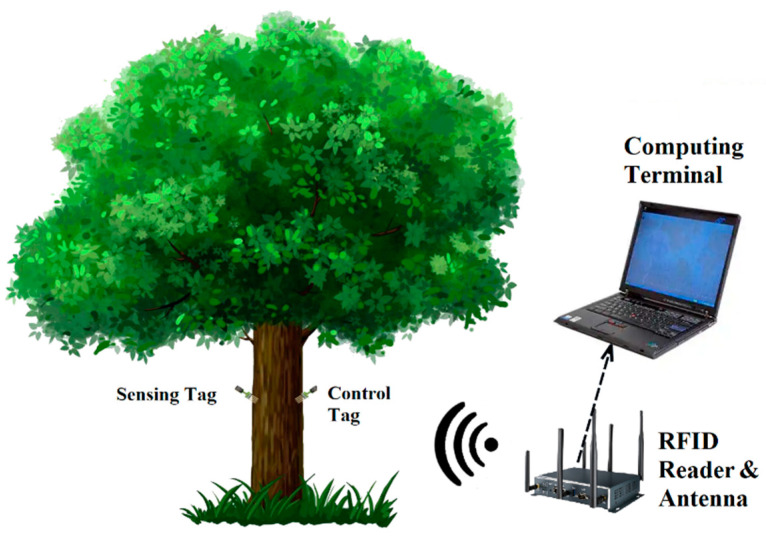
Structure of the moisture content monitoring system. The system includes an UHF RFID reader and antenna, computing terminal, multiple sets of measuring tags, and different measured woods.

**Figure 2 sensors-22-06287-f002:**
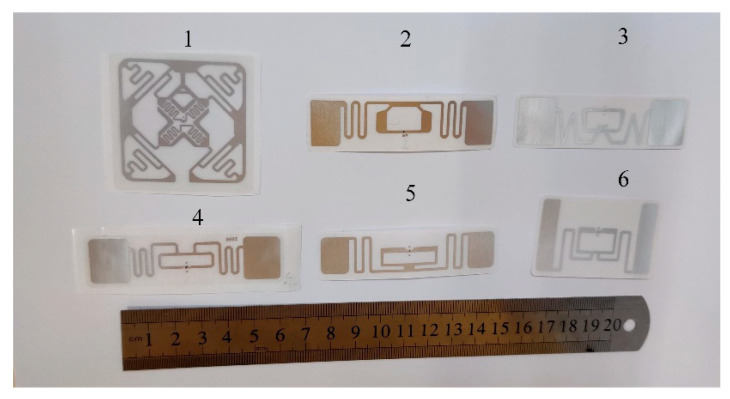
UHF RFID tags selected in the experiment (labeled 1–3 from left to right on the top line, and 4–6 from left to right at the bottom).

**Figure 3 sensors-22-06287-f003:**
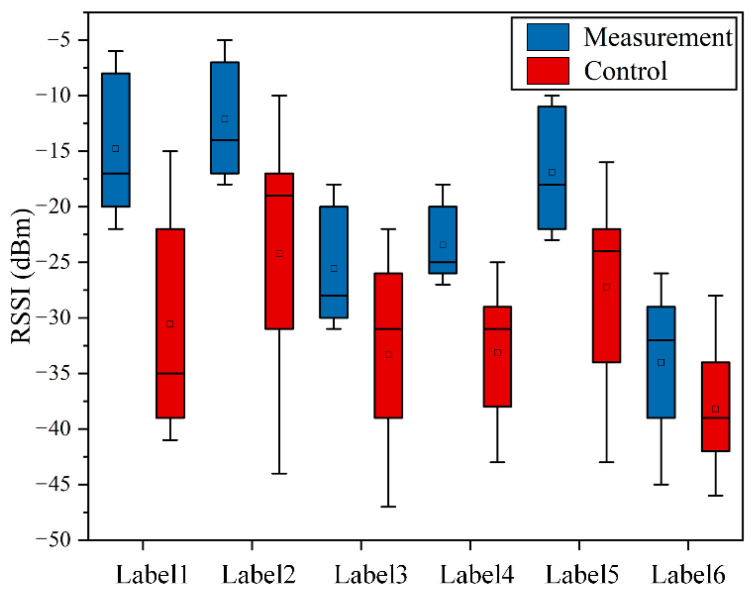
The signal strength that different types of tag can transmit within one meter (tested on a boxwood sample with 17% MC).

**Figure 4 sensors-22-06287-f004:**
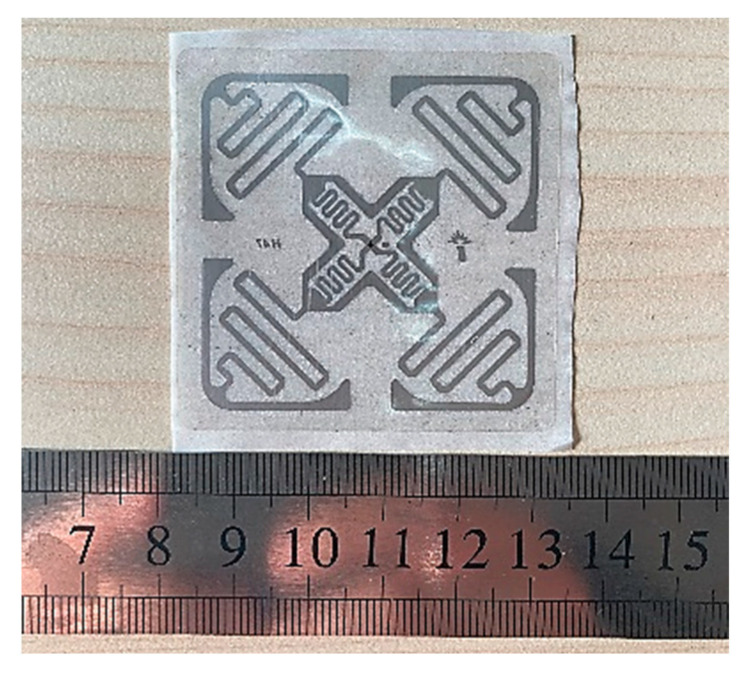
UHF RFID Inlay Tag (JT-IH47); the length and width of the label are both 5.12 cm.

**Figure 5 sensors-22-06287-f005:**
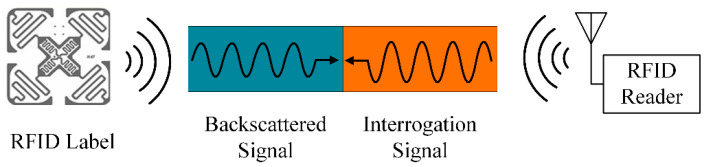
The sensing principle of UHF RFID: the information exchange between the tag and the reader is achieved through the interrogation signal and the backscattered signal.

**Figure 6 sensors-22-06287-f006:**
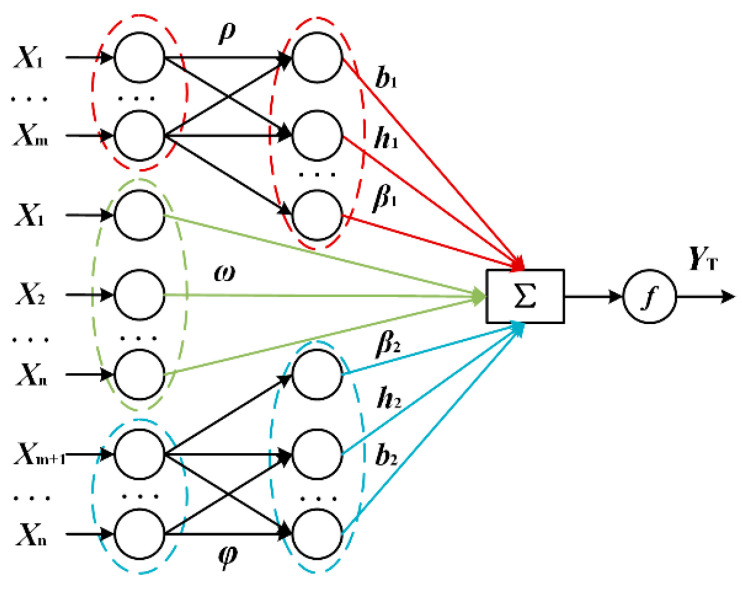
The OS-PELM structure. A strategy of multiple hidden layers is adopted to optimize the network structure.

**Figure 7 sensors-22-06287-f007:**
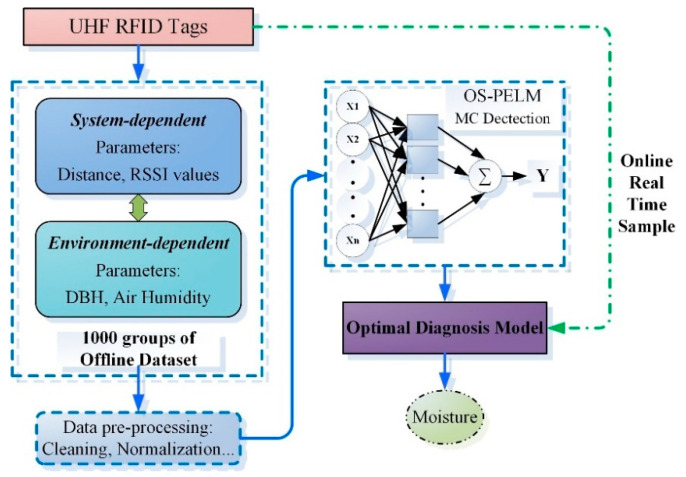
Framework diagram of the MC prediction model.

**Figure 8 sensors-22-06287-f008:**
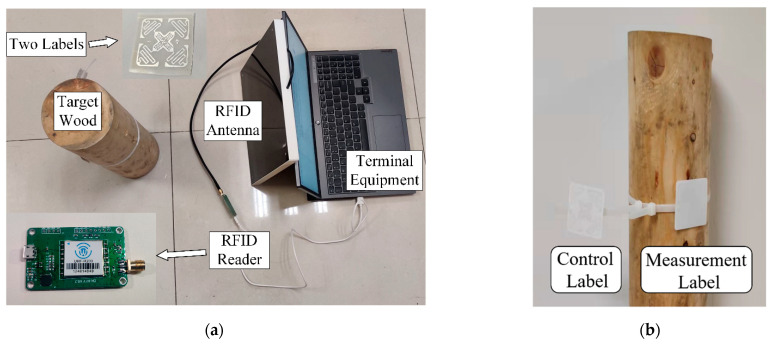
Laboratory experimental scenario with the boxwood sample. (**a**) shows the setup for the actual measurement in the laboratory, and (**b**) shows an example of the sticker attached to the tree.

**Figure 9 sensors-22-06287-f009:**
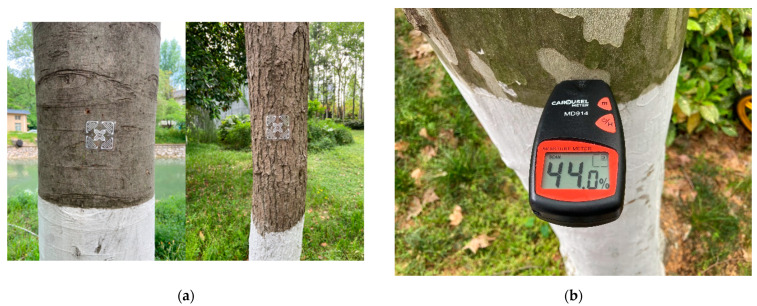
Measurements on standing trees. (**a**) shows the schematic diagram of sticking labels on the standing trees, and (**b**) shows the actual moisture content of trees calibrated with MD914.

**Figure 10 sensors-22-06287-f010:**
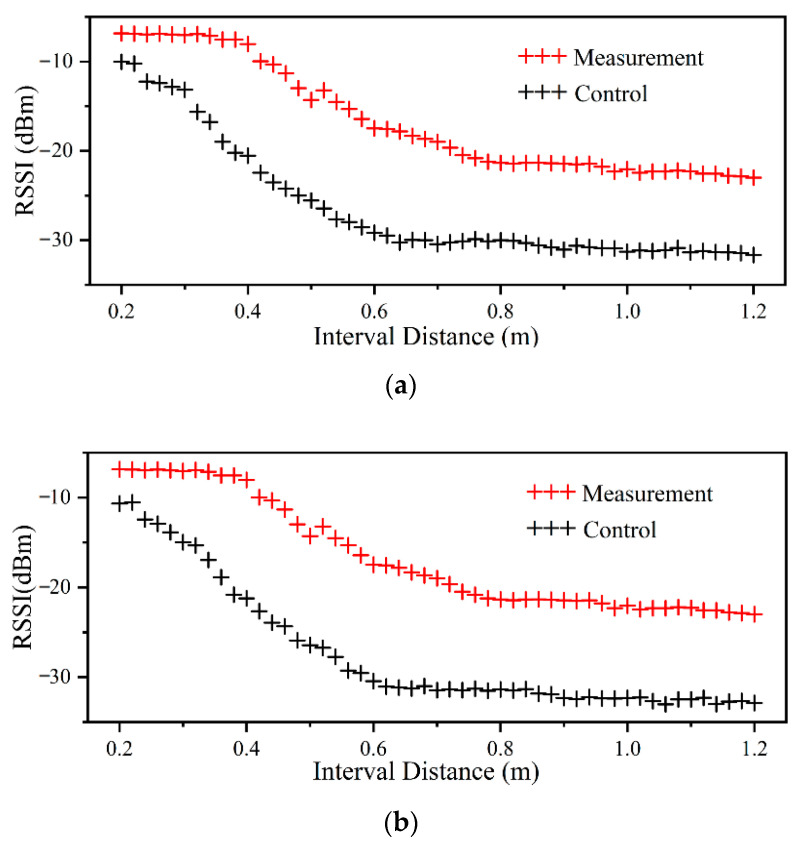
Laboratory RSSI values with samples containing different MC levels. (**a**) is a laboratory-prepared wood with a 35% MC level, and (**b**) is a laboratory-prepared wood with a 25% MC level.

**Figure 11 sensors-22-06287-f011:**
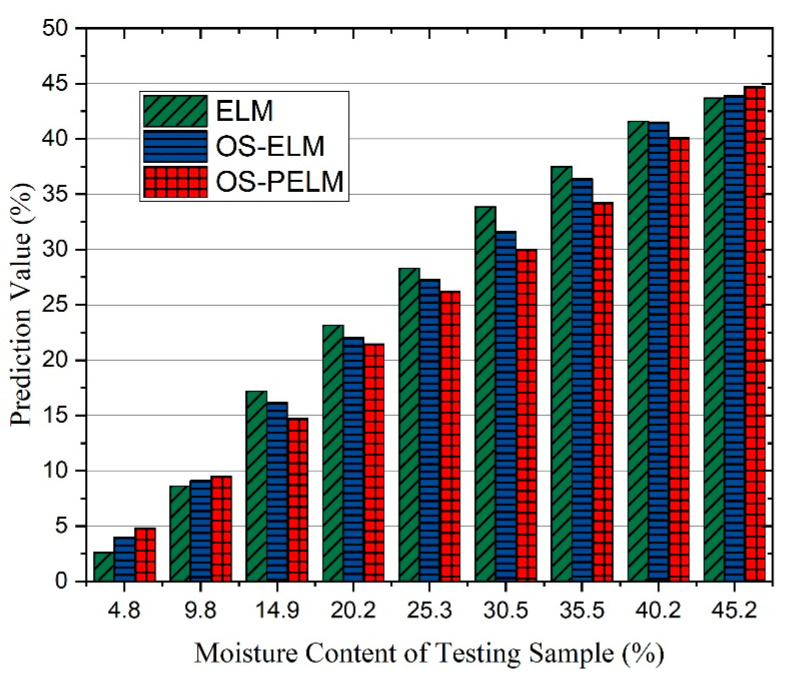
The prediction results of the OS-PELM algorithm and two comparison algorithms on nine sample woods with different MC levels.

**Figure 12 sensors-22-06287-f012:**
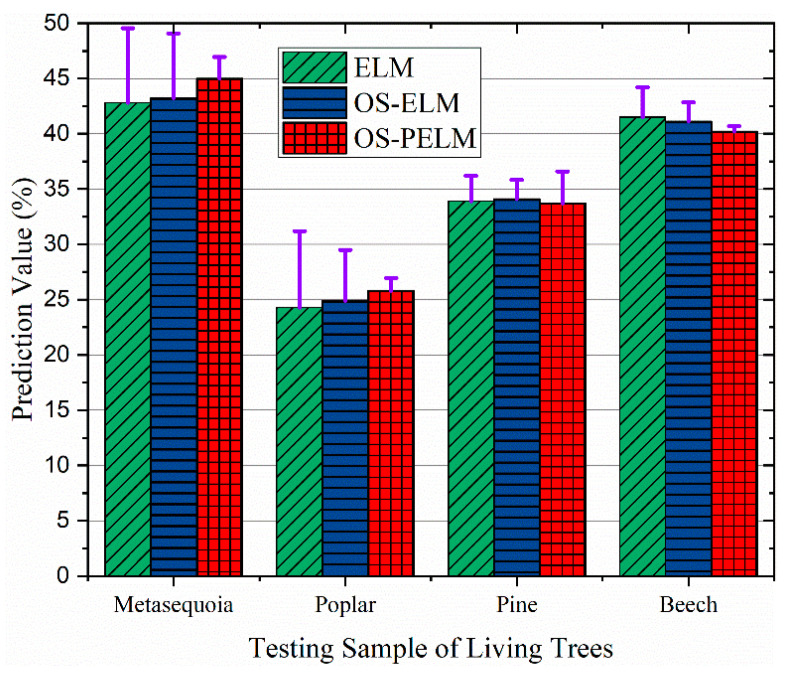
The prediction results of the OS-PELM algorithm and two comparison algorithms for four different kinds of living trees (metasequoia, poplar, pine and beech, with MC levels of 45.9%, 26.1%, 34.7%, and 40.4%).

**Figure 13 sensors-22-06287-f013:**
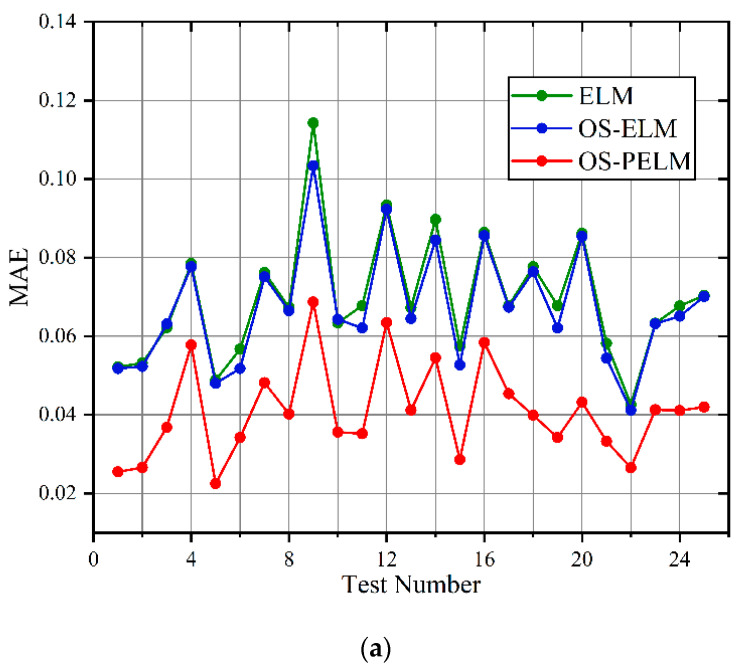
Prediction error evaluation of the three models. (**a**) is the MAE of the three models, (**b**) is the RMSE of the three models, and (**c**) is the R^2^ of the three models.

**Figure 14 sensors-22-06287-f014:**
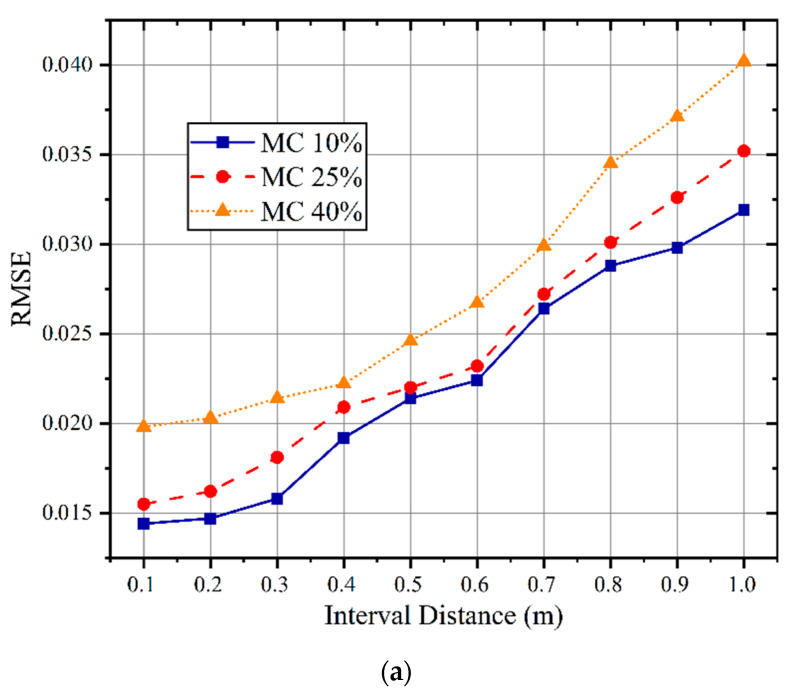
The interval distance vs. RMSE in (**a**) standing cypress and (**b**) the laboratory.

**Figure 15 sensors-22-06287-f015:**
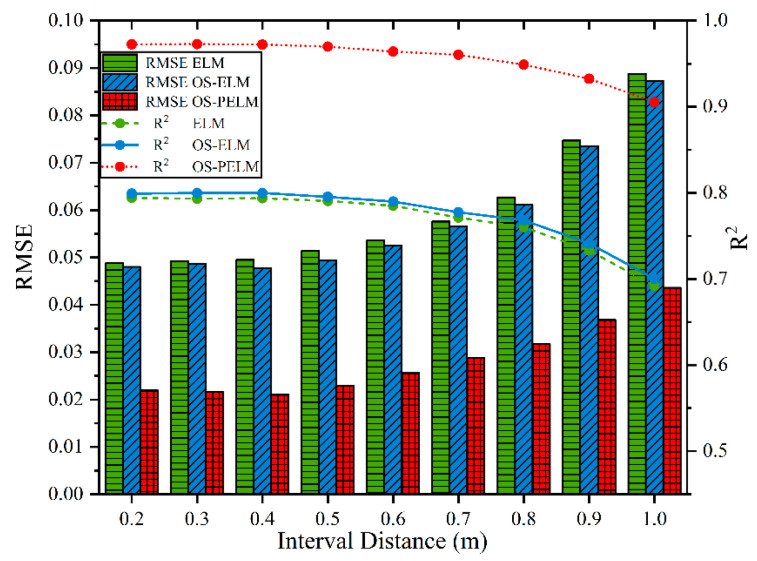
The interval distance vs. the performance indicators of the three models.

**Figure 16 sensors-22-06287-f016:**
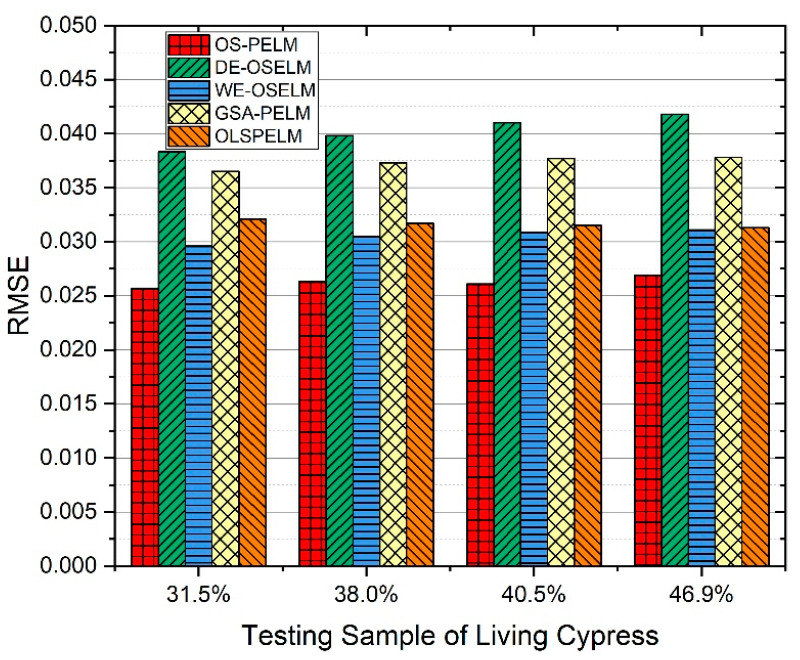
For four samples of cypress trees with different moisture contents (31.5%, 38.0%, 40.5%, and 46.9%), the prediction accuracy of the five methods was evaluated.

**Table 1 sensors-22-06287-t001:** Parameters related to differential backscatter power in UHF RFID sensing.

Symbol	Description
*P_r_*	Backscattered power measured by the reader
*P_tag-back_*	Backscattered power of the tag
*P_tx_*	Reader antenna transmission power
*G_tag_*	Tag antenna gain
*G_tx_*	Reader antenna gain
*τ*	Antenna–chip impedance matching coefficient
*d*	Distance between antenna and tag
*λ*	RF wavelength
∆	Variation parameters of physical functions

**Table 2 sensors-22-06287-t002:** Components of typical measuring data.

Wood Moisture Content (%)	Measuring Distance (m)	Air Humidity (%)	DBH (m)	Measuring RSSI (Average of dBm)
14.8	0.3	54	0.16	−10.05
20.5	0.3	54	0.16	−12.12
25.6	0.3	54	0.16	−13.21
29.8	0.4	55	0.24	−17.63
40.8	0.4	55	0.24	−18.88
40.8	0.5	55	0.24	−24.06
40.8	0.6	55	0.24	−27.34
15.3	0.3	53	0.27	−8.45
20.2	0.3	53	0.27	−10.12
25.2	0.3	53	0.27	−10.98

**Table 3 sensors-22-06287-t003:** Predictive performances of MC models.

Algorithm Model	MAE	RMSE	R^2^	Runtime (s)
ELM	0.0488	0.0565	0.8086	7.21
OS-ELM	0.0480	0.0554	0.8155	7.88
OS-PELM	0.0225	0.0254	0.9612	7.96

## Data Availability

All data generated or presented in this study are available upon request from the corresponding author. Furthermore, the models and code used during the study cannot be shared at this time as the data also form part of an ongoing study.

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
