# Peer review of "Living Tree Moisture Content Detection Method Based on Intelligent UHF RFID Sensors and OS-PELM"

_sensors, 2022, doi:10.3390/s22166287_

Round 1
Reviewer 1 Report
In this manuscript, the authors introduced the use of UHF RFID and OS-PELM sensors to detect the moisture content of living trees. The work was presented clearly and the originality and the novelty of the work were well defined. The work is interesting and has a number of potential applications.
The writing of the manuscript needs some improvement especially the language. I would suggest the authors to proofread the whole manuscript.
Reviewer 2 Report
The Authors have submitted the manuscript on extent research in the field of smart UHF RFID sensors and feedforward neural network algorithms used to determine the moisture content of living trees. The paper covers all components of a job well prepared and performed. At the beginning, work motivation and main aims have been precisely defined. Basing on the provided state-of-the-art in the field of living tree moisture content detection, RFID technique and the possibility of applying RSSI signal fluctuations in prediction models, the proposed range of the research has been strongly justified. According to the invented method, the Authors have adjusted multi aspect experiment for measuring and predicting wood moisture, both in laboratory site and in real conditions of a forestry farm. They conducted a well-thought-out analysis and in-depth discussion of the results. They have also compared their methods with studies carried out by other researchers. So I judge the paper as well composed and it can be published as is. Of course it is possible to find a minor understatements in the text, but this does not influence on the value of the manuscript. Also, a slight linguistic correction could be required. I come up some mistakes such us “generaliz-ability”, “composi-tion”, “Non-destructive”; strange sentences e.g. in lines 90-92, 122,…; wrong style of the letters of variables, notation of the values and their units; wrong numeration of Sections in lines 84-87, inappropriate division of table and figure and their captions between two pages (similarly for Algorithm 1), etc.
Finally, I would like to ask only one case about the RSSI value of the backscattered signal from sensing tag. As we know, the signal from the RFID reader is subjected to multipath propagation. I wonder if the reflected signals could disturb the prediction of the moisture content (especially for experiment conducted in the laboratory).
Reviewer 3 Report
Dear Editor in Chief of Sensors
As the reviewer of the manuscript entitled as ‘Living Tree Moisture Content Detection Method Based on In-telligent UHF RFID Sensors and OS-PELM I went through the manuscript and found that it has merits to publish in an international Journal. It suffers from minor shortcomings that I have highlighted on the attached PDF manuscript file. In overall, I am recommending for publication after minor revision. Moreover, I am strongly suggesting to be improved linguistically and grammatically by a native in English.
Sincerely Yours .
